# Nonlinear eco-evolutionary games with global environmental fluctuations and local environmental feedbacks

Yishen Jiang[1,3], Xin Wang[2,3,4,5,6,7]\*, Longzhao Liu[2,3,4,5,6,7], Ming Wei[1,3], Jingwu Zhao[8], Zhiming Zheng[2,3,4,5,6,7,9,10], Shaoting Tang[2,3,4,5,6,7,9,10]\*

**1** School of Mathematical Sciences, Beihang University, Beijing, China, **2** Institute of Artificial Intelligence, Beihang University, Beijing, China, **3** Key laboratory of Mathematics, Informatics and Behavioral Semantics (LMIB), Beihang University, Beijing, China, **4** State Key Lab of Software Development Environment (NLSDE), Beihang University, Beijing, China, **5** Zhongguancun Laboratory, Beijing, P.R.China, **6** Beijing Advanced Innovation Center for Future Blockchain and Privacy Computing, Beihang University, Beijing, China, **7** PengCheng Laboratory, Shenzhen, China, **8** School of Law, Beihang University, Beijing, China, **9** Institute of Medical Artificial Intelligence, Binzhou Medical University, Yantai, China, **10** School of Mathematical Sciences, Dalian University of Technology, Dalian, China

\* wangxin_1993@buaa.edu.cn (XW); tangshaoting@buaa.edu.cn (ST)

**Data Availability Statement:** Codes are available at https://github.com/lileason/PCOMPBIOL-D-22-01825.git.

## Abstract

Environmental changes play a critical role in determining the evolution of social dilemmas in many natural or social systems. Generally, the environmental changes include two prominent aspects: the global time-dependent fluctuations and the local strategy-dependent feedbacks. However, the impacts of these two types of environmental changes have only been studied separately, a complete picture of the environmental effects exerted by the combination of these two aspects remains unclear. Here we develop a theoretical framework that integrates group strategic behaviors with their general dynamic environments, where the global environmental fluctuations are associated with a nonlinear factor in public goods game and the local environmental feedbacks are described by the 'eco-evolutionary game'. We show how the coupled dynamics of local game-environment evolution differ in static and dynamic global environments. In particular, we find the emergence of cyclic evolution of group cooperation and local environment, which forms an interior irregular loop in the phase plane, depending on the relative changing speed of both global and local environments compared to the strategic change. Further, we observe that this cyclic evolution disappears and transforms into an interior stable equilibrium when the global environment is frequency-dependent. Our results provide important insights into how diverse evolutionary outcomes could emerge from the nonlinear interactions between strategies and the changing environments.

## Author summary

The intricate interplay between strategic behavior and environment is ubiquitous in complex systems of different scales. Previous works mainly focus on one aspect of the

**Funding:** This work was supported by National Key R&D Program of China Grant 2022ZD0116800 (ST), and Program of National Natural Science Foundation of China under Grant No. 62141605 (ZZ), 12201026 (XW), 11922102 (ST), 11871004 (ST). The funders had no role in study design, data collection and analysis, decision to publish, or preparation of the manuscript.

**Competing interests:** The authors have declared that no competing interests exist.

environmental changes: either global environment fluctuations that unidirectionally decide the welfare of the evolutionary dynamics, or local environment feedbacks that coevolve with the strategic behavior. Here we develop a theoretical framework that integrates them both in order to obtain a more complete picture of how group cooperation evolves in a general dynamic environment. We show that global environmental fluctuations can fundamentally alter the dynamical predictions of local game-environment evolution. The most interesting finding is the emergence of cyclic evolution of group cooperation and local environment, which forms an interior irregular loop in the phase plane, depending on the relative changing speed of both global and local environments compared to the strategic change. Such irregular loop, however, is substituted by an interior stable fixed point when considering a more complicated situation where the global environment is also frequency-dependent. Our results show how rich dynamical outcomes arise from the interactions between strategic behaviors and their natural or social environments, which has important practical value for solving social dilemmas in an ever-changing world.

## Introduction

Cooperation promotes the emergence of stronger adaptabilities and more abundant functions in many species, forming the very basis of natural systems at different scales [1–4]. However, the 'selfish gene' widely exists in all biosystems where individuals always make rational choices based on their own benefits [5–7], giving rise to social dilemmas of non-cooperation [8]. Naturally, understanding why persistent cooperation occurs ubiquitously, under what conditions stable cooperation could be maintained or promoted, and how the cooperative behavior evolves under natural selections has long been the core objective of evolutionary game theory [9–12]. In particular, many different mechanisms have been proposed to address the well-known Prisoner's Dilemma and the Tragedy of the Commons [13, 14], for instance, kin selection [15], direct reciprocity and indirect reciprocity [16–18], punishment and reward [19–21], spatial reciprocity [22–24] and group selection [25]. More realistically, the heterogeneity of players is also taken into account, such as network topology [26], selective participation mechanism [10, 27], wealth-based selection [28], and the recently studied higher-order interactions [29].

While the early evolutionary game approach typically focuses on the internal properties of replicator dynamics [9], assuming that the strategic interactions happen in a fixed environment, the impact of a dynamic environment is ignored. Therefore, coevolutionary games incorporating an exogenous environment's evolution process have been largely studied [10, 30]. Coevolution rules introduce the environment-related characteristics into the game, for instance, the interaction network, the size of the population, the mobility, aging, and reputation of players, which also evolve in time and could affect the evolutionary outcome of strategies [31]. Further, the ecological factors in microbial systems are abstracted into a global time-dependent environment [32]. Such global environmental changes, reflecting the periodic ecological fluctuations or the rapid ecological perturbations, modify the payoffs of the game through a time-varying function and show highly complex impacts on the evolution of group cooperation and the evolutionary balance of phenotypes [32, 33].

Note that coevolutionary games only consider the feedback from the global environment, the dynamics of which is independent of the strategic interactions. However, the bi-directional

feedbacks between strategies and the environment are identified in a wide range of real-world systems [34–36]. In microbial systems, cooperation often arises due to the secretion or the release of extracellular enzymes, extracellular antibiotic compounds, and growth factors, which modifies the local environmental state that will, in turn, alter the incentive for public goods production [2, 4, 37, 38]. Likewise, in modern society, decision-making dynamics of competitive cognitions can reshape the public opinion environment, especially with the rise of large-scale social networks, and this shared media atmosphere in turn affects the benefit of social discussion in the decision-making process [39]. In fact, the interactions between online public discourse and the external political environment can lead to the emergence of polarized echo chambers [40], which has aroused great concern in recent years [41]. Similar coupled dynamics can also be obtained in psychological-economic systems, social-ecological systems, and human-medicine systems [36, 38, 42, 43], relating to a number of big challenges such as global climate change, overfishing, anti-vaccine problems, pandemic prevention and control, cancer treatment [44–48].

To characterize such complex feedback loops, an emerging theory of 'eco-evolutionary games' is proposed recently [49, 50]. In eco-evolutionary game, the strategic behaviors change the state of the environment, while in turn, the environment alters the payoff structure of the game, driving the replicator dynamics with a strategy-dependent feedback-evolving game [51]. Abundant evolutionary outcomes are observed under such framework. Beginning from the simplest form, a two-player game coupled with linear environmental feedback can already generate the persistent oscillation of both population cooperations and environmental states [52]. Similar persistent cycles can also occur in asymmetric games with heterogeneous environments [53–55]. As a meaningful extension, a multi-player game coupled with asymmetrical environmental feedback identifies the threshold of the feedback speed that can yield oscillatory convergence to persistent cooperation, highlighting the importance of time-scales [56]. An innovative manifold control approach is further proposed to steer the eco-evolutionary dynamics to a desired direction [57].

Despite the progress, current models consider only one aspect of the dynamic environments, either global environmental changes which are time-dependent or local environmental feedbacks which are strategy-dependent. In real-world systems, however, these two aspects often coexist and exert complex forces on strategic evolution. For example, in the context of the COVID-19 pandemic, cooperation in public health measures has strong impacts on disease spreading, and vice versa [47]. Beyond this eco-evolution, the seasonal fluctuations of the virus's transmissibility also alter the payoffs of the strategic behaviors [58]. In crowdsourcing projects, cooperation can emerge from the asymmetric incentive feedback, resulting in a local feedback loop [56, 59]. Meanwhile, the periodic fluctuations of the global economic environment [60], which obviously cannot be influenced by the strategic behaviors within the projects, could affect the synergy and discounting of the group payoffs [61]. In microbial systems, the feedback loop between bacterial evolution and the local environment could also arise due to the existence of asymmetric preferential access to public goods, while the changes in culture environment, serving as the exogenous global environment, could remarkably affect their strategic behavior [37, 62]. Understanding the evolution of cooperation in such complex systems thus has profound practical significance, calling for a framework that could unveil the complete picture of the environmental effects where both global and local environmental dynamics are incorporated.

Here we develop a theoretical framework that integrates group strategic behaviors with their general dynamic environments. In specific, the global environmental fluctuation that influences the group benefit is characterized by a nonlinear factor in public goods game, while the local environmental feedback driven by an asymmetric incentive mechanism is

described by the 'eco-evolutionary game'. Of particular interest, we show how global environmental fluctuations alter the dynamical predictions of the group cooperation in local feedback-evolving game. Different from the previous observation that local game-environment dynamics could eventually evolve to a stable interior fixed point where cooperation and defection coexist, we find the emergence of cyclic evolution of group cooperation and local environment under a periodically changing global environment. Surprisingly, such eco-evolutionary dynamics could form an interior closed yet irregular orbit in the phase plane, depending on the relative time-scale of both global and local environments versus strategic changes. However, when the global environment is not only time-dependent but also frequency-dependent, such interior closed orbit is substituted by an interior stable fixed point. Our results provide novel insights toward how complex group behaviors emerge from the nonlinear interactions between strategies and dynamic environments, especially in a non-autonomous system, which is important for understanding the evolution of social dilemmas in a changing world.

## Methods

In order to study the evolution of group cooperation, which is ubiquitous in microbial systems and in human society, we consider a modified nonlinear public goods game (PGG) among a well-mixed infinitely large population. In every game, each of the $N$ participants randomly drawn from the infinite population can choose to be a cooperator contributing $c$ to the public pool, or a defector reaping without sowing. In classic PGG, after the total contribution is multiplied by the multiplication factor $r$, the total benefit is equally distributed to each participant. However, it has been obtained that group cooperation often emerges due to the existence of preferential access to the valuable common good or other extra incentives for cooperators [56, 57], which brings about asymmetric payoff structures for cooperators and defectors and further drives the local feedback-evolving game. Here we distinguish multiplication factors of cooperators and defectors as $r_c$ and $r_d$, respectively. In addition, the actual benefits provided by cooperators may depend nonlinearly on the number of cooperators and on the total investments, the former is common in biology and the latter has been largely revealed in economics [63]. Hence, we adopt the modeling idea proposed in [61] and capture such nonlinearity by a nonlinear factor $w$.

   Accordingly, the payoffs for each defector and cooperator in a group with $k$ cooperators, $P_d(k)$ and $P_c(k)$, are

$$P_d(k) = \frac{r_d c}{N}\left(1 + w + w^2 + \cdots + w^{k-1}\right) = \frac{r_d c}{N}\frac{1 - w^k}{1 - w}, \tag{1}$$

$$P_c(k) = \frac{r_c c}{N}\left(1 + w + w^2 + \cdots + w^{k-1}\right) - c = \frac{r_c c}{N}\frac{1 - w^k}{1 - w} - c, \tag{2}$$

such that the benefits created by each additional cooperator are rescaled, either synergistically enhanced when $w > 1$, or discounted when $w < 1$. For instance, as the concentration of enzymes produced by cooperators increases, the enzyme-mediated reaction exhibits a faster rate than linearity, indicating that additional enzyme production has enhanced payoffs ($w > 1$). On the contrary, the benefit provided by the first cooperator in foraging yeast cells may be critical for survival, whereas the value of additional food decreases ($w < 1$) [61]. Therefore, $P_d(k)$ and $P_c(k)$ are convex when $w > 1$ and are concave when $w < 1$. Finally, when $w = 1$, the classic linear PGG can be recovered. In a population with a fraction $x$ of cooperators, for

any focal individual, the probability that $k$ out of $N-1$ other participants are cooperators is

$$\binom{N-1}{k} x^k (1-x)^{N-1-k},\tag{3}$$

The average fitness of defectors and cooperators, $f_d$ and $f_c$, are thus given by

$$\begin{aligned}f_d &= \sum_{k=0}^{N-1}\binom{N-1}{k} x^k (1-x)^{N-1-k} P_d(k)\\ &= \frac{r_d c}{N(1-w)}\left[1-(1-x+wx)^{N-1}\right],\end{aligned}\tag{4}$$

$$\begin{aligned}f_c &= \sum_{k=0}^{N-1}\binom{N-1}{k} x^k (1-x)^{N-1-k} P_c(k+1)\\ &= c\left\{\frac{r_c}{N(1-w)}\left[1-w(1-x+wx)^{N-1}\right]-1\right\}.\end{aligned}\tag{5}$$

For simplicity and without loss of generality, we specify that the contribution of each cooperator is 1. The changes in the fraction of cooperation over time, namely the evolution of group cooperation, is then described by the replicator dynamics:

$$\begin{aligned}\dot{x} &= x(f_c-\bar{f}) = x(1-x)(f_c-f_d)\\ &= x(1-x)\left(\frac{r_c\left(w(wx-x+1)^{N-1}-1\right)}{N(w-1)}-1-\frac{r_d\left((wx-x+1)^{N-1}-1\right)}{N(w-1)}\right),\end{aligned}\tag{6}$$

where $\bar{f} = xf_c + (1-x)f_d$ is the average fitness of the population.

Further, we introduce two prominent aspects of the environmental influence into the framework: the global environmental fluctuations and the local environmental feedbacks. Generally, the global environment significantly affects the total and marginal benefits of group cooperation. For instance, companies tend to increase the salary and recruit more employees in bull markets, adopting aggressive expansion strategies as each additional staff could provide much more returns, while on the contrary, they are more likely to cut salaries and reduce the stuff in bear markets. Similarly, changes in nutrient concentration of culture medium or the usage of drugs directly affect the fitness of the competing bacteria or cancer cells [37, 38, 62, 64]. Such global environmental fluctuations could naturally be reflected and characterized by the nonlinear factor $w$: a large synergy effect of cooperation corresponds to a good global environment where $w > 1$, while the discounting of cooperation is related to a bad global environment where $w < 1$. Therefore, we simply use a time-dependent function $w = w(t)$ to depict changes of the global environment over time.

In addition, the local environmental feedback arised from the asymmetric incentive mechanism, which is strategy-dependent, is characterized by the dynamics of cooperator's multiplication factor $r_c$ following [56]:

$$\dot{r}_c = \epsilon(r_c-\alpha)(\beta-r_c)f(x,r_c),\tag{7}$$

where $\epsilon > 0$ denotes the relative changing speed of $r_c$ compared with $x$. Due to limited resources in the local environment, $r_c$ is confined to the range $[\alpha, \beta]$ and we have $1 < \alpha < \beta < N$ according to the social dilemma in PGG. Moreover, $f(x,r_c)$ describes the asymmetric feedback mechanism in the model, whose sign determines the increase or decrease in $r_c$. To mimic

the fact that in many microbial systems, the preferential access mechanism affects the welfare distribution of cooperators and defectors and that cooperation is first favored and then constrained due to the limitation of total resources and the zero-sum feature of resource consumption, we assume $f(x, r_c)$ is a linear function of cooperators' and defectors' payoffs following the idea in [56]

$$f(x, r_c) = -xf_c + \theta(1-x)f_d, \tag{8}$$

where $xf_c$ and $(1-x)f_d$ are the expected payoffs for cooperators and defectors, respectively. $\theta > 0$ denotes the distribution ratio of the expected total payoff of the cooperators and defectors. Such local feedback-evolving dynamics could facilitate cooperation by increasing $r_c$ when $x$ is small, while in turn, a relatively large $x$ leads to the decrease of the cooperator's rewards, subjecting to the law of diminishing marginal utility. Besides, to reduce model complexity, the defectors' multiplication factor $r_d$ is set to be constant. We assume $r_d \leq r_c$ to describe the widespread phenomenon that cooperators have preferential access to the common good in microbial and social systems [56, 57, 65, 66]. For the sake of simplicity, we set $r_d \leq \alpha \leq r_c$ throughout the paper.

The complete modeling framework is illustrated by Fig 1. The dynamics of our nonlinear eco-evolutionary game with global environmental fluctuations and local environmental feedbacks, which describes the complex group strategic behaviors in general dynamic environments, can thus be written as

$$\begin{cases} \dot{x} = x(1-x)\left( \dfrac{r_c(w(t)(w(t)x - x + 1)^{N-1} - 1)}{N(w(t) - 1)} - 1 - \dfrac{r_d((w(t)x - x + 1)^{N-1} - 1)}{N(w(t) - 1)} \right) \\[2ex] \dot{r}_c = \epsilon(r_c - \alpha)(\beta - r_c)\left( -x\left( \dfrac{r_c(w(t)(w(t)x - x + 1)^{N-1} - 1)}{N(w(t) - 1)} - 1 \right) \right. \\[2ex] \left. + \dfrac{r_d\theta(1-x)((w(t)x - x + 1)^{N-1} - 1)}{N(w(t) - 1)} \right) \end{cases} \tag{9}$$

## Results

### Nonlinear dynamics of local game-environment evolution in static global environments

We first study the nonlinear dynamics of local game-environment evolution under static global environments where $w(t)$ is a fixed constant. Correspondingly, the eco-evolutionary game described by Eq 9 degenerates into an autonomous system. There are totally seven possible fixed points in the system, six of which are on the boundary and the remaining one is an interior equilibrium point (see detailed proof of the stability of all seven fixed points in S1 Appendix). Only two boundary fixed points are possible to be stable: (i) $(x^* = 0)$ is always stable, leading to a full defection among the population which also occurs in the classic PGG. (ii) $(x^* = 1, r_c^* = \alpha)$ is stable only if $\frac{\alpha(w^N - 1)}{N(w-1)} > 1$ and $r_d$ is smaller than the threshold $r_d^* = \frac{\alpha(w^N - 1) - N(w-1)}{(w^{N-1} - 1)}$. Under this circumstance, the system evolves to full cooperation with a minimum value $\alpha$ of the cooperators' multiplication factor. The phase diagram of the stability of this fixed point with respect to $w$ and $r_d$ is shown in Fig 2a. Not surprisingly, the full cooperation situation tends to emerge from a healthy global environment with larger $w$ and smaller $r_d$.

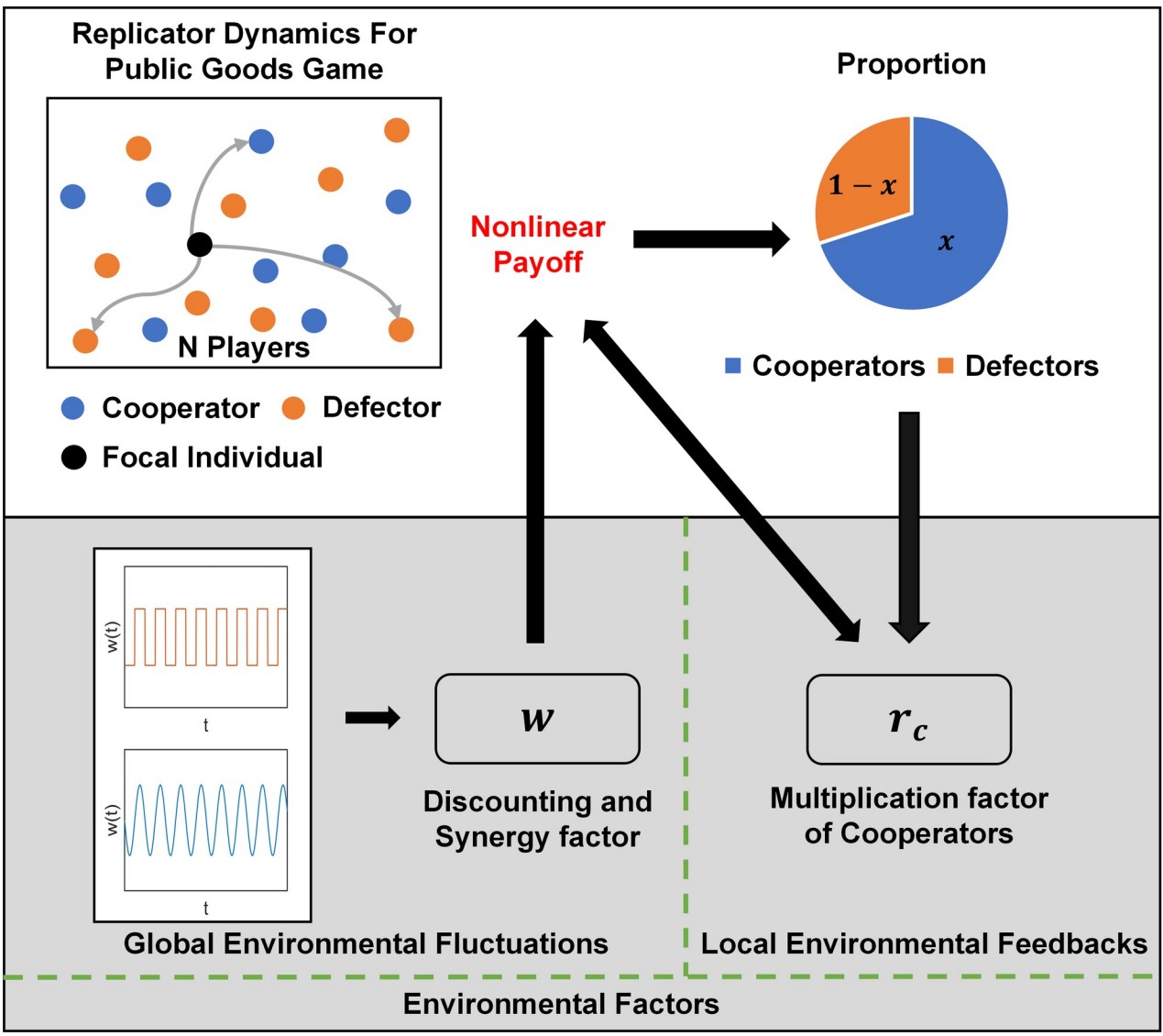

**Fig 1. Schematic of the eco-evolutionary games with general dynamic environments.** (Top) The group strategic behaviors are described by a nonlinear evolutionary public goods game. (Top, Bottom) The influence of environmental changes consists of two prominent aspects: the global environmental fluctuations that directly affect the synergy and discounting of the group payoffs, characterized by the nonlinear factor $w$, and the asymmetric environmental feedbacks that drive the local strategy-dependent feedback-evolving game, characterized by the multiplication factor of cooperators $r_c$.

Of particular interest, we analyze under what condition the interior equilibrium point

$$\left( x^* = \frac{\theta}{\theta + 1}, r_c^* = \frac{N(w-1) + r_d\left((wx^* - x^* + 1)^{N-1} - 1\right)}{w(wx^* - x^* + 1)^{N-1} - 1} \right),\tag{10}$$

could be stable, bringing about the coexistence of cooperators and defectors with an intermediate local environmental state. Since $\alpha < r_c^* < \beta$, we first identify the existence condition of

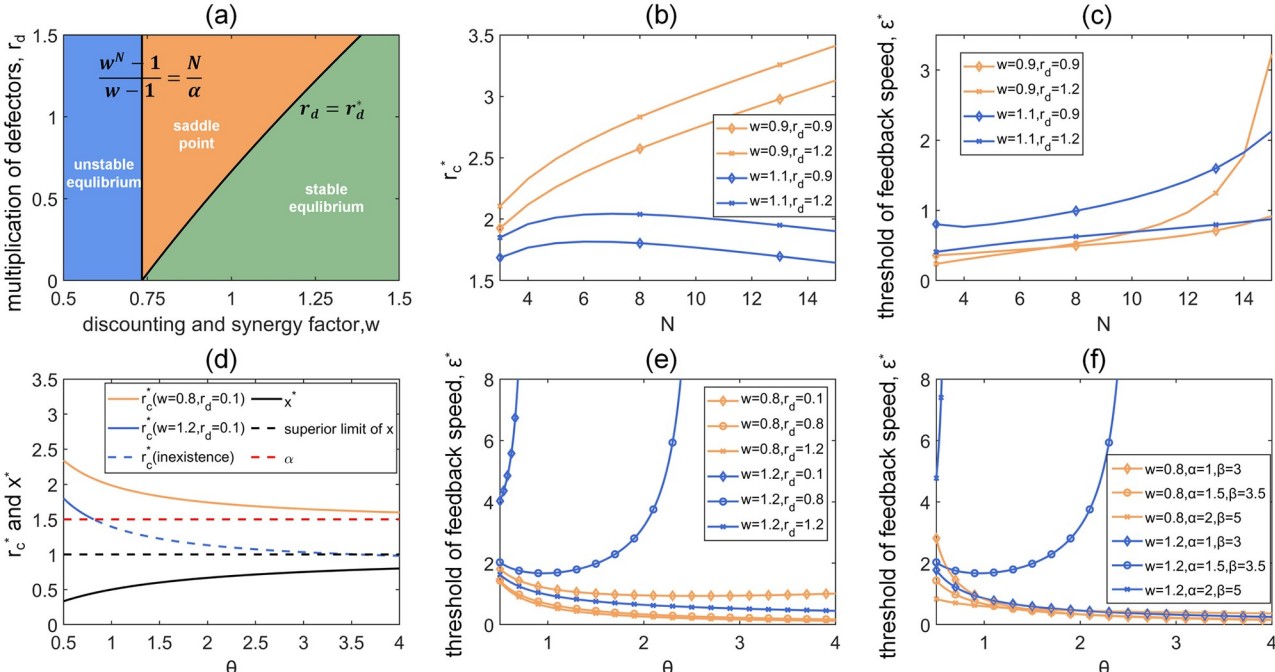

**Fig 2. Effects of varying modeling parameters on equilibrium states of local game-environment evolution in static global environments.** Panel (a) shows the combined influence of $w$ and $r_d$ on the stability of the boundary fixed point $(1, \alpha)$. Panel (b) and (c) show the detailed influence of group size $N$ on the equilibrium states revealed by $r_c^*$ and $\epsilon^*$, respectively. Panel (d) presents the trends of the interior fixed point $(x^*, r_c^*)$ as $\theta$ changes (the dashed blue line indicates that the value of $r_c^*$ is out of parameter range $[\alpha, \beta]$). Panel (e) and (f) show how the threshold of the relative feedback speed $\epsilon^*$, which is the minimum value leading to the interior stability, is affected by different parameters. Parameters: $N = 4$ in (a, d-f), $\alpha = 1.5$, $\beta = 3.5$ in (a, c, e) and $r_d = 0.8$ in (f).

this equilibrium point

$$
\max\left\{0, \frac{\alpha\left(w\left(\frac{w\theta+1}{\theta+1}\right)^{N-1} - 1\right) - N(w-1)}{\left(\frac{w\theta+1}{\theta+1}\right)^{N-1} - 1}\right\} \leq r_d
$$

$$
\leq \min\left\{\alpha, \frac{\beta\left(w\left(\frac{w\theta+1}{\theta+1}\right)^{N-1} - 1\right) - N(w-1)}{\left(\frac{w\theta+1}{\theta+1}\right)^{N-1} - 1}\right\}.
$$

(11)

Furthermore, the existing interior fixed point is stable only if the relative feedback speed of the local environment $\epsilon$ exceeds a threshold $\epsilon^*$, which can be written as follows:

$$
\epsilon^* = \frac{(r_c^*w - r_d)(N-1)(w\theta+1)^{N-2}(w-1)}{(r_c^* - \alpha)(\beta - r_c^*)\left[w(w\theta+1)^{N-1} - (1+\theta)^{N-1}\right]}.
$$

(12)

Then, we show how the group size $N$ influences the equilibrium states of local game-environment evolution revealed by $r_c^*$ and $\epsilon^*$, respectively (Fig 2b and 2c). To ensure that the interior fixed point always exists as $N$ changes, we set $w = 0.9$ and 1.1, $r_d = 0.9$ and 1.2, $\theta = 2$. We find that in a discounting global environment, $r_c^*$ is approximately in direct proportion to $N$, while in the synergy condition, $r_c^*$ increases first and then decreases gradually as $N$ increases

(Fig 2b). Interestingly, $r_c^*$ is always larger when $w$ is smaller or $r_d$ is larger regardless of the changes in $N$. This is intuitive: when the global environment is worse or the multiplication factor of defectors is larger, the payoff coefficient of cooperators should be increased in order to maintain the stable coexistence of cooperators and defectors. We also find that $\epsilon^*$ increases as $N$ increases under all four sets of parameters, indicating that the emergence of the stable interior fixed point requires faster local environmental feedback when the group size becomes larger (Fig 2c).

As shown by Eq 10, the final frequency of cooperator $x^*$ is solely determined by $\theta$, the distribution ratio of cooperator's and defector's total payoffs. As $\theta$ increases, $x^*$ also increases whereas the stable multiplication factor of cooperators $r_c^*$ decreases (Fig 2d). Such changing trends are intuitive in many teamworks: an unskilled project, which has a relatively larger $\theta$, often requires more participants with lower benefits. On the contrary, in a skilled project such as scientific collaboration, the decrease of $\theta$ results in small research teams with higher rewards for each member.

Further, Fig 2e shows how $\epsilon^*$ varies as $w$, $r_d$ and $\theta$ change. In a less optimistic global environment where the group benefits are discounted ($w = 0.8$), $\epsilon^*$ gently decreases as $\theta$ increases. In a booming global environment where the group benefits are synergistically enhanced ($w = 1.2$), however, the trends become complicated, depending on the multiplication factor of defectors $r_d$. When $r_d$ becomes smaller, $\epsilon^*$ sharply increases as the interior fixed point tends to disappear, the latter is shown by the dash line in Fig 2d. Similar trends can also be observed under different combinations of $\alpha$ and $\beta$ (Fig 2f).

In Fig 3, we show phase dynamics of the local game-environment evolution under different static global environments. We set $w = 0.8, 1, 1.2$ to mimic the scenarios of discounting, linear, and synergy PGG in each column, respectively. Throughout the paper, we fix $N = 4$, $\alpha = 1.5$, $\beta = 3.5$. Other parameters are $\theta = 2$ and $r_d = 1$. Accordingly, we can calculate the corresponding threshold $\epsilon^*$ using Eq 12 and select parameters $\epsilon$ that are smaller than, equal to or larger than $\epsilon^*$, respectively. Consistent with our theoretical predictions, the persistent coexistence of cooperators and defectors only occurs when $\epsilon > \epsilon^*$ in all three scenarios, where the system oscillatorily converges to the interior equilibrium state. In particular, group cooperation may only arise from the local asymmetrical environmental feedbacks that are quick and timely enough, especially when the global environment is relatively poor (Fig 3a, 3d and 3g and Fig 3b, 3e and 3h). Moreover, we find that a good global environment significantly promotes the emergence of full cooperation. An important insight is that a slower local environmental feedback, though obstructs the emergence of interior equilibrium, could indeed increase the basin of attraction of full cooperation (Fig 3c, 3f and 3i). In addition, as the benefits brought by the global environment $w$ increase, the stable coexistence of cooperation and defection arise with less asymmetric incentive feedback $r_c^*$, leading to a reduction in the basin of attraction of the interior equilibrium. Our results show a non-negligible role and complicated joint influence of global environmental state and local environmental feedback on group strategy evolution.

## Nonlinear dynamics of local game-environment evolution in dynamic global environments

Further, we study how dynamic global environments affect the local game-environment evolution. Specifically, we focus on periodic global environmental fluctuations, which are typical and widespread in various complex systems. For instance, the daily cycle of sunlight, the seasonal fluctuations of ecological characteristics, the cyclical economic crisis, etc. Here we consider one type of periodic change: discrete shifts modeled by a piecewise function $w_1(t)$, which

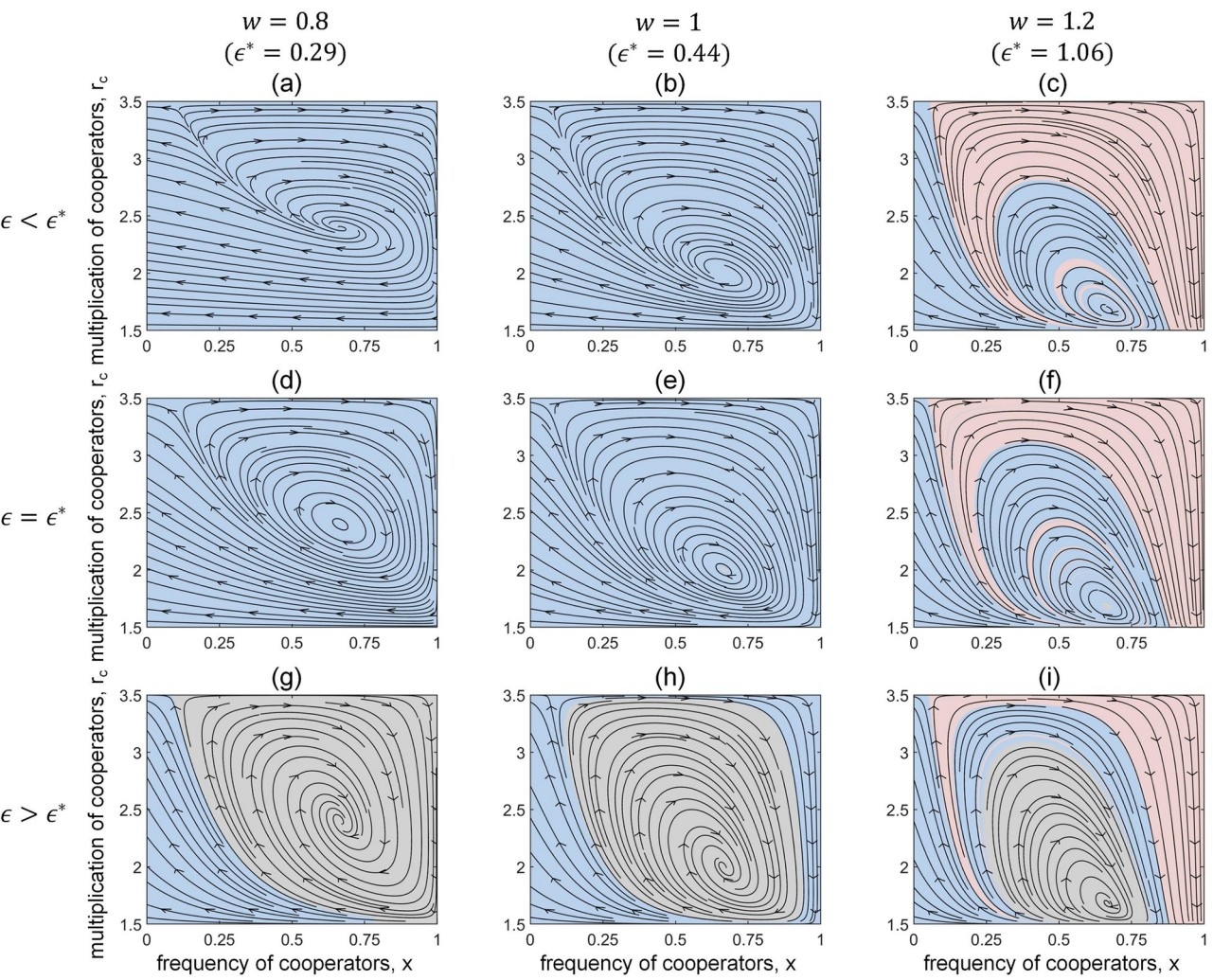

**Fig 3. Phase plane dynamics of local game-environment $(x - r_c)$ evolution with $N = 4$, $\alpha = 1.5$, $\beta = 3.5$, $\theta = 2$, $r_d = 1$.** We set $w = 0.8, 1, 1.2$ to describe the scenarios of discounting, linear, and synergy PGG in each column, and the corresponding $\epsilon^*$ are 0.29, 0.44, 1.06, respectively. We therefore choose $\epsilon = 0.09, 0.24, 0.86$ for the first row such that $\epsilon < \epsilon^*$, and $\epsilon = 0.79, 0.94, 1.56$ for the third row such that $\epsilon > \epsilon^*$. The blue, pink and gray areas represent the basin of attraction of different fixed points $(x^* = 0)$, $(x^* = 1, r_c^* = \alpha)$ and $(x^* = \frac{\theta}{\theta+1}, r_c^* = \frac{N(w-1)+r_d((wx^*-x^*+1)^{N-1}-1)}{w(wx^*-x^*+1)^{N-1}-1})$, respectively. The stable coexistence of cooperators and defectors only occurs when $\epsilon > \epsilon^*$ (the third row), as predicted theoretically, while a synergistically enhanced global environment significantly promotes the emergence of full cooperation (the third column).

is simply given by

$$
w_1(t) = \begin{cases} 1.3 & \left[\frac{t}{T}\right] = 2n \\ \\ 0.7 & \left[\frac{t}{T}\right] = 2n + 1 \end{cases}, n = 0, 1, 2, \cdots\cdots. \tag{13}
$$

where $T$ is half period of $w_1(t)$. For the convenience of comparison to the continuous case where $w = w_3(t) = 1 - 0.5 \sin (at + \delta)$ as shown in S2 Appendix, we let $T = \frac{\pi}{a}$ and set $w_1(t)$ change between $\frac{1}{T} \int_0^T (1 - 0.5 \sin at) dt \approx 0.7$ and $\frac{1}{T} \int_T^{2T} (1 - 0.5 \sin at) dt \approx 1.3$, where $a$ decides the time scales of global environmental fluctuations (see S2 Appendix).

In Fig 4, we show the dynamical trajectories of local game-environment evolution under a discretely varying global environment $w = w_1(t)$. The initial points are uniformly selected on the phase plane and the trajectories are calculated numerically by Eq 9. We fix $\theta = 0.5$ and $r_d = 0.6$. The thresholds of the local environmental feedback speed corresponding to the two values in $w_1(t)$ thus can be obtained using Eq 12, which are $\epsilon_1^* = 1.45$ and $\epsilon_2^* = 2.86$. Therefore, we choose $\epsilon = 1, 2.5, 6$, which satisfies $1 < \epsilon_1^* < 2.5 < \epsilon_2^* < 6$, such that our analysis could contain all the possible situations. Furthermore, considering the fact that the global environmental changes are commonly much slower than the strategy evolution, we set the relative changing speed $a = 0.01, 0.1, 1$ (see S2 Appendix) and the corresponding periods are $2T = 200\pi$, $20\pi$ and $2\pi$, respectively. We find that the global environmental fluctuations could fundamentally alter the dynamical predictions of the group cooperation in local feedback-evolving games. When $\epsilon = 1$, the local eco-evolutionary dynamics evolve either to full defection where $x = 0$ (blue

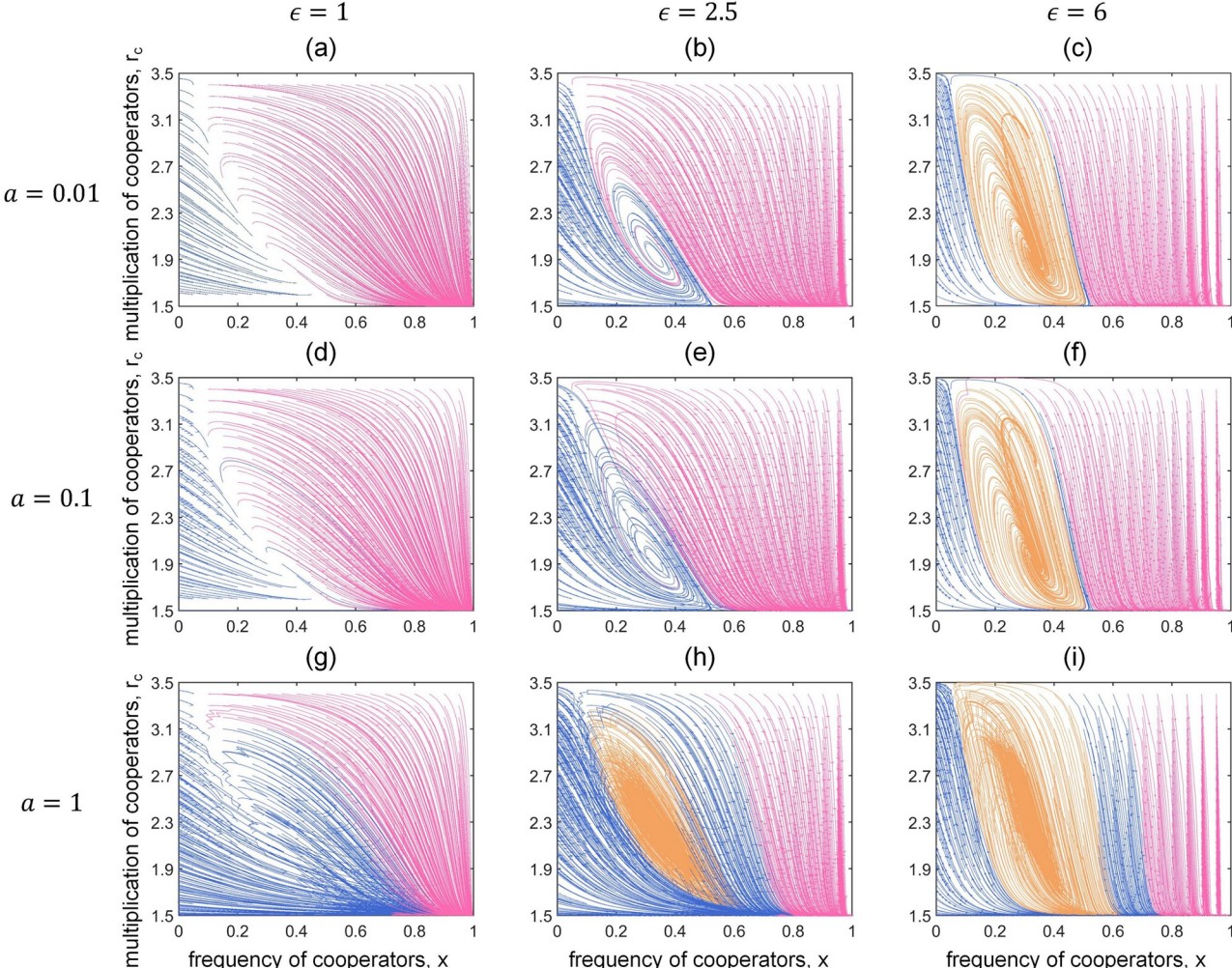

**Fig 4. Local game-environment evolution in a discretely varying global environment.** We use a periodic piecewise function, $w_1(t)$, to describe the global environmental fluctuations. We uniformly select initial points on the $x - r_c$ plane and plot the corresponding dynamical trajectories by numerically solving Eq 9. Trajectories that eventually evolve to $(x^* = 0)$, $(x^* = 1, r_c^* = \alpha)$ or circulate along an interior closed orbit are distinguished by blue, pink and orange, respectively. The emergence of cyclic evolution of group cooperation and local environment (orange areas), which cannot be observed in static global environments, indicates that the global environmental fluctuations could fundamentally alter the evolutionary outcomes in local feedback-evolving game. In all panels, $N = 4$, $\alpha = 1.5$, $\beta = 3.5$, $\theta = 0.5$, $r_d = 0.6$.

trajectories) or to full cooperation where $x = 1$ and $r_c = \alpha$ (pink trajectories). When $\epsilon = 2.5$, however, a new evolutionary outcome emerges when the global environment changes fast: the local game-environment dynamics will eventually circulate along an interior closed orbit (orange trajectories). More specifically, different from the phenomenon that the system oscillatorily converges to the interior equilibrium as we observed in static global environments, here we find that a periodically changing global environment could lead to the emergence of cyclic evolution of group cooperation and the local environment. When $\epsilon = 6$, such cyclic evolution can even emerge from various time-scales of global environmental changes. Similar evolutionary trends can also be observed in a continuously changing global environment (see S2 Appendix).

These results just show how diverse evolutionary group behaviors emerge from the complex interactions between strategies and general dynamic environments, especially highlighting the important role of relative time-scales of both global and local environments versus strategic changes ($T$ and $\epsilon$). Then, we concentrate on the newly discovered phenomenon that the local game-environment dynamics can evolve cyclically (Fig 5). Due to the complexity of the non-autonomous system and the lack of theoretical insight, we display in detail the phase plane dynamics of local game-environment evolution, the typical dynamic trajectories with different initial conditions, and the corresponding time evolution of the frequency of cooperators $x$ and the multiplication factor of cooperators $r_c$, in order to provide a clear view of the cyclic evolution. We set $\theta = 1$, $r_d = 1.2$, $T = 10\pi$ and choose $\epsilon = 4$ which is larger than the maximum threshold of $\epsilon^*$ with regard to $w_1(t)$. Importantly, we find that the reason for the formation of cyclic evolution under discrete global environment $w_1(t)$ is that the two fixed points are in the mutual attraction domains (Fig 5b). In the shown cases, the initial points first evolve to the stable fixed point $(x^* = 0.5, r_c^* = 1.87)$ where $w = 1.3$ within time $T$. Subsequently, the global environment changes and fixed point 1 becomes a new initial point, which is captured by another stable fixed point $(x^* = 0.5, r_c^* = 2.92)$ corresponding to $w = 0.7$. Similarly, in the next $T$ time, the evolution is directed to fixed point 1 again, resulting in the formation of an interior closed orbit. The periodic fluctuations of group cooperation and the local environment are shown in Fig 5c and 5d. Such periodic evolution is actually in line with the intuitions from the real world in a way that the group cooperation will neither disappear completely nor always be maintained at the highest level in many complex systems, for instance, the seasonal oscillating dynamics of the COVID-19 infection [67–70], the dynamic adjustment of big companies in the economic cycles and the oscillating abundance of bacteria or cancer cells in periodically varying environments [37, 64].

In addition, we confirm whether the emergence of cyclic evolution is robust with regard to the changes of group size $N$ and the distribution ratio of the expected total payoffs of cooperators and defectors $\theta$. We find that the group size $N$ could significantly affect the range of the orange region. Specifically, larger group size may hinder the emergence of cyclic evolution (Fig 6a and 6b), making it more difficult to reach the dynamic coexistence of cooperators and defectors. We also explore the influence of $\theta$. Fig 6c and 6d show that the final position of the interior closed orbit moves as $\theta$ changes, owing to the influence of $\theta$ on the fixed points of local game-environment evolution. The robustness of other parameters (such as the multiplication factor of defectors $r_d$ and the range of cooperators' multiplication factor $[\alpha, \beta]$), and the sensitivity analysis of the continuous model are also considered (see S3 Appendix).

For now, the global environment is only assumed to be time-dependent. In real-world complex systems, the global environment could certainly exhibit more complicated manners. As a heuristic case study, we further examine how the evolutionary outcomes are influenced when the global environment is also frequency-dependent and is in a threshold-function manner,

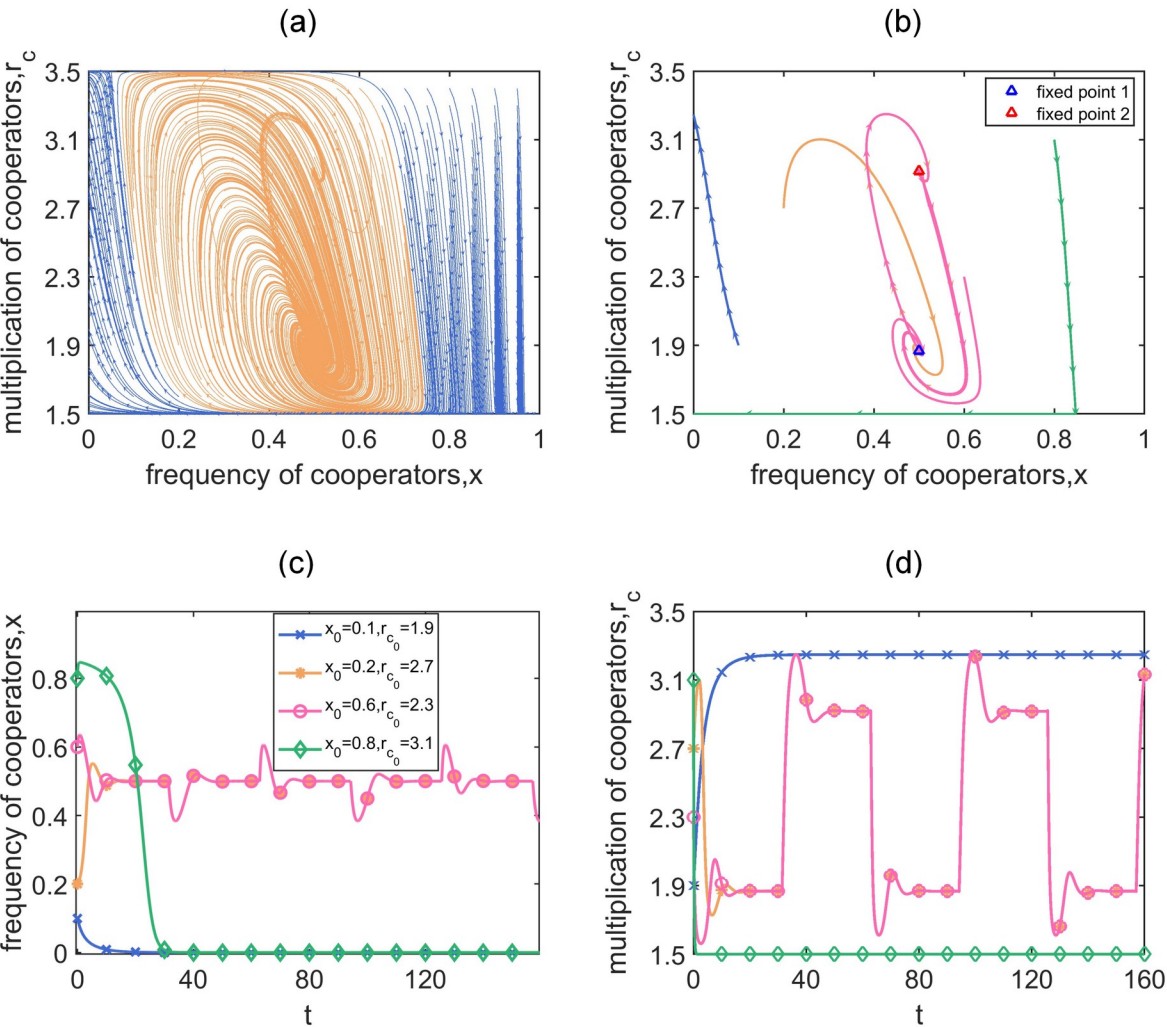

**Fig 5. Emergence of cyclic evolution of group cooperation and local environment under periodically changing global environments given by $w_1(t)$.** Panel (a) shows local game-environment evolution similar to Fig 4. Panel (b) displays four typical dynamic trajectories in detail, particularly the interior closed yet irregular orbits. The last row presents time evolution of the frequency of cooperators $x$ and the multiplication factor of cooperators $r_c$ under different initial conditions, corresponding to the colored trajectories in panel (b). Results show that the formation of cyclic evolution under discrete global environment is due to the fact that the two fixed points are in the mutual attraction domains (b-d). Parameters: $N = 4$, $\alpha = 1.5$, $\beta = 3.5$, $\theta = 1$, $r_d = 1.2$, $T = 10\pi$, $\epsilon = 4$.

which is denoted by $w_2(t)$:

$$w_2(t) = \begin{cases} 1.3, x(t) < K \\ 0.7, x(t) \geq K \end{cases}, \qquad (14)$$

where $K$ is a threshold value between 0 and 1. When the frequency of cooperators $x(t)$ is smaller than the threshold $K$, the global environment becomes synergistically enhanced in order to facilitate cooperation, while on the contrary, when $x(t)$ is larger than $K$, the global environment becomes discounted. Fig 7 shows local game-environment evolution under these frequency-dependent global environments. Surprisingly, we find that unlike the time-dependent cases, the internal periodic orbit disappears, instead, an interior stable equilibrium

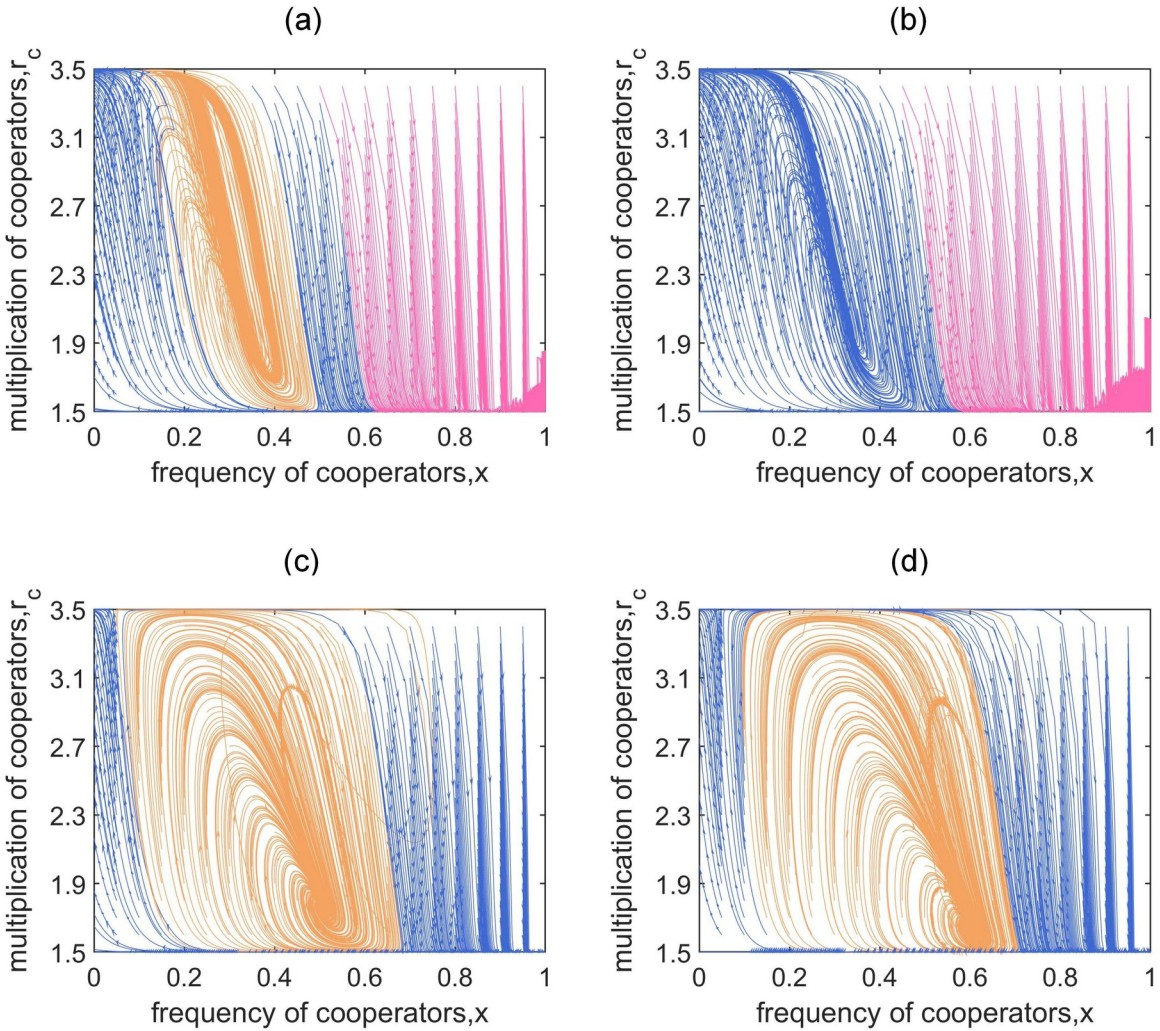

**Fig 6. Game-environment evolution under different sizes of group $N$ and distribution ratio of the expected total payoffs of the cooperators and defectors $\theta$.** Trajectories on $x - r_c$ phase plane eventually evolve to $x^* = 0$, $(x^* = 1, r_c^* = \alpha)$ or circulate along an interior closed orbit, which are distinguished by blue, pink and orange, respectively. Panels (a) and (b) correspond to $N = 5$ and 6 with $\theta = 0.5$, $r_d = 0.6$, $T = 2\pi$ and $\epsilon = 9$, while panels (c) and (d) correspond to $\theta = 1$ and 1.5 with $N = 4$, $r_d = 1$, $T = 10\pi$ and $\epsilon = 6$. In all panels, $\alpha = 1.5$, $\beta = 3.5$.

(where the grey trajectories stabilize in Fig 7a–7c) emerges under different thresholds. We further present the detailed evolutionary trajectories of several representative initial points in Fig 7d–7f. The phase plane is divided into two regions by the threshold $K$, with orange on the left representing $w(t) = 1.3$ and blue on the right representing $w(t) = 0.7$. The triangles represent the stable interior fixed points, and the vertical dotted lines indicate $x = \frac{\theta}{\theta+1}$, which is also the abscissa of the interior fixed points. Obviously $x = \frac{\theta}{\theta+1}$ can only fail into one of the two regions, either orange or blue, and the corresponding $w(t)$ decides the final position of the interior stable equilibrium. In other words, the system can only have one interior fixed point in our threshold model. It is also worthy of note that a stable boundary equilibrium may emerge when the threshold $K$ is large (Fig 7c and 7f).

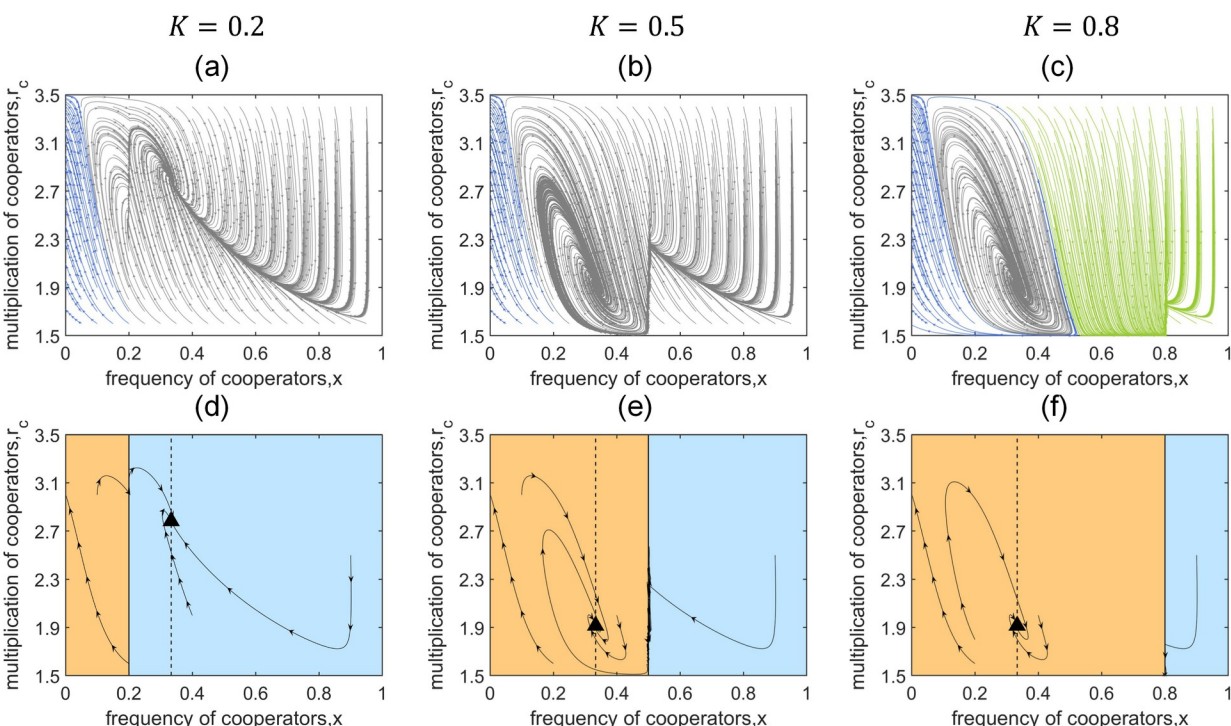

**Fig 7. Local game-environment evolution under frequency-dependent global environment.** We assume the global environment is in a simple threshold-function manner denoted by $w_4(t)$. In specific, when the frequency of cooperators $x(t)$ is smaller than the threshold $K$, the global environment becomes synergistically enhanced in order to facilitate cooperation, while on the contrary, when $x(t)$ is larger than $K$, the global environment becomes discounted. The first row shows the overall evolutionary trends. Trajectories that eventually evolve to full defection, the stable interior fixed point and the boundary equilibrium point on the x-axis are distinguished by blue, grey and green, respectively. The second row shows the detailed evolutionary trajectories of several representative initial points in the phase plane. The plane is divided into two regions by the threshold K, with orange on the left representing $w(t) = 1.3$ and blue on the right representing $w(t) = 0.7$. The vertical dotted and solid lines represent $x = \frac{\theta}{\theta+1}$ and $x = K$, respectively, and the triangle represents the stable interior fixed point. Unlike the time-dependent cases, the internal periodic orbit disappears, instead, an interior stable equilibrium emerges under all thresholds. For comparison, the parameters are the same as in Fig 4: $N = 4, \alpha = 1.5, \beta = 3.5, \theta = 0.5, r_d = 0.6, \epsilon = 6$.

## Discussion

How rich dynamical outcomes arise from the interactions between strategic behaviors and their natural or social environments is one of the fundamental questions in many complex systems across disciplines. On the one hand, the global environment unidirectionally changes the total welfare of the evolutionary dynamics. On the other hand, the local environment provides frequency-dependent feedback that modifies the payoff structure of the game dynamics, while in turn the strategies taken by individuals can also reshape the state of the local environment over time. Particularly, two important theoretical frameworks that describe the latter feedback loop are proposed: the stochastic game [71, 72], and the eco-evolutionary game [52, 55–57]. Stochastic games introduce game transition mechanisms to depict the discrete changes of the external environments, i.e., the cooperation behavior in the current game can affect the game that individuals play in the next period. Meanwhile, eco-evolutionary game theory characterizes the continuous environmental changes coupled with strategic interactions via a set of ordinary differential equations. However, previous models exclusively focus on one aspect of environmental changes. To obtain a more complete picture of the group behavioral evolution in a general dynamic environment, we develop a modeling framework that integrates the more

complicated influence exerted by both global environmental fluctuations and local environmental feedbacks.

Real interactions between strategic behaviors and their environments are commonly nonlinear. Our analysis shows how this nonlinearity, relating to the state of the global environment and the marginal benefits provided by the cooperators, affects the local game-environment evolution. We find that in a static global environment, regardless of the scenarios of discounting, linear, and synergy, the persistent coexistence of cooperation and defection only emerges if the relative feedback speed of the local environment exceeds a certain threshold, breaking the 'Tragedy of the Commons'. The nonlinear factor, however, could determine the occurrence of full cooperation and influence the attraction basin of the stable interior equilibrium.

As the influence of global dynamic environments on the local eco-evolutionary game is the primary focus of our model, our results emphatically show how the periodic global environmental fluctuations fundamentally alter the evolutionary outcomes of group cooperation. The most intriguing finding is the emergence of an interior closed yet irregular orbit in the local game-environment phase plane, leading to the cyclic evolution of group cooperation and local environment, which is firstly discovered in the multi-player situation and qualitatively in line with the oscillating dynamics in two-player games [52, 55]. We unveil that this new dynamical phenomenon can be intuitively understood as a limit of the continuously converging process of the dynamical paths confined by the fixed points. Importantly, our theoretical framework has shown the crucial role of relative time-scales of both global and local environments compared to strategic interactions. Further, when examining a more intricate scenario where the global environment is also frequency-dependent, we find an interior stable equilibrium instead of cyclic evolution emerges, underscoring the significance of the nature of the global environment.

To sum up, our model provides profound insight into how diverse group behaviors, especially oscillating convergence and cyclic evolution, can emerge from the nonlinear interactions between strategies and dynamic environments. Our work helps understand the complexity of social dilemmas in an ever-changing world. For instance, the game dynamics of the Labor and Capital in the social-economic system can be easily translated into our frame: The companies engaging staff can be mapped into the local environment, the state of which coevolves with individual strategies [73]. Outside, the trends of the corresponding industry or the economic state of the country, which can largely determine the marginal benefits of strategic behaviors, can be modeled as the global environment. Such an approach allows us to examine how the evolutionary stable state of the labor-capital game is jointly influenced by factors such as the incentive mechanisms of the company and the development of the economies. Our results also have important practical value in systems biology. Studies have shown that the global periodically varying environment associated with biotic (blood circulation, nutrients supply) or abiotic (periodical drug usage) variations could induce nonlinear competition between different types of tumor cells in local environments, which drastically changes the evolutionary outcomes and results in oscillating dynamics of tumor population, breaking the dominance of a certain type of tumor cell [64]. Our model incorporating nonlinear local interactions as well as general mechanisms of global environmental changes thus paves ways for a more confined understanding on such dynamical processes of cancer evolution. For instance, the parameters in global environmental changes could be exploited for designing milder control strategies for tumor growth. Similarly, in microbial systems, the concentration of casein, corresponding to the global environment, could lead to an oscillation coexistence of copiotroph (W04) and oligotroph (Y09) in the local environment [37]. In microbial experiments, the dilution factor in the growth medium significantly influences the local feedback loop between the laboratory yeast population and the dynamics of cooperators, the SUC2 gene [62]. Our model can be

used to describe these complex microbial systems and may help in establishing effective control methods that could tune the system evolution to a desired direction.

Though in this work we only consider the periodic fluctuations and a simple threshold-function manner of the global environment which are widespread in natural systems, various changing rules can be studied by simply giving different time-evolving functions $w(t)$. Future works may further consider the structured interactions with increasing complexity and the control strategies in the framework of game-environment dynamics.

## Supporting information

**S1 Appendix. Stability of fixed points in static global environments.**
(PDF)

**S2 Appendix. Discretely varying and continuously changing global environments.** We first consider a continuous function to describe the continuously changing global environment, which exhibits similar evolutionary trends to Fig 4. Then we consider a four-segment function to validate our explanations about the emergence of interior closed orbit in continuously changing global environments.
(PDF)

**S3 Appendix. Sensitive analysis on modeling parameters.** We conduct sensitive analysis on all parameters including the multiplication factor of defectors $r_d$, the group size $N$, the distribution ratio of the expected total payoffs of cooperators and defectors $\theta$, the range of cooperators' multiplication factor $[\alpha, \beta]$ and the initial phase $\delta$.
(PDF)

## Author Contributions

**Conceptualization:** Yishen Jiang, Xin Wang.

**Funding acquisition:** Xin Wang, Zhiming Zheng, Shaoting Tang.

**Investigation:** Yishen Jiang, Xin Wang, Longzhao Liu, Ming Wei, Jingwu Zhao.

**Supervision:** Xin Wang, Zhiming Zheng, Shaoting Tang.

**Validation:** Yishen Jiang, Xin Wang, Longzhao Liu.

**Writing – original draft:** Yishen Jiang.

**Writing – review & editing:** Xin Wang, Longzhao Liu, Shaoting Tang.

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
