## [Decision Letter · Decision Letter 0]

7 Feb 2023

Dear Mr. Wang,

Thank you very much for submitting your manuscript "Nonlinear eco-evolutionary games with global environmental fluctuations and local environmental feedbacks" for consideration at PLOS Computational Biology.

As with all papers reviewed by the journal, your manuscript was reviewed by members of the editorial board and by several independent reviewers. In light of the reviews (below this email), we would like to invite the resubmission of a significantly-revised version that takes into account the reviewers' comments.

We cannot make any decision about publication until we have seen the revised manuscript and your response to the reviewers' comments. Your revised manuscript is also likely to be sent to reviewers for further evaluation.

Sincerely,

Christian Hilbe

Academic Editor

PLOS Computational Biology

James O'Dwyer

Section Editor

PLOS Computational Biology

Comments by the Academic Editor: 

The article has been read by three expert reviewers.

They all appreciate the proposed framework, in which exogenous and endogenous factors jointly affect the game-environment dynamics.

However, they also all make a number of constructive suggestions to further improve the paper. In particular, they stress that the writing should be improved and that the impact of different model parameters should be discussed in more detail.

I concur with all these suggestions. Please take them into account when preparing a revised manuscript.

In addition, please make sure that all data and computational code is made publicly available. Currently, the suggested github repository seems to be empty.

Reviewer's Responses to Questions

**Comments to the Authors:**

Reviewer #1: The authors in this paper combine global environmental fluctuations and local environmental feedbacks in a human-environment system. They consider individuals playing a public goods game where more than one cooperator is present in the interactions. The global fluctuations can be depicted by introducing the synergy and discounting of cooperation (a nonlinear factor omega). On the other hand, the local feedbacks can be captured by coupling the individual behavior and the local game environment (how the multiplication factor r change in the public goods game for cooperators).

Given that everything takes place in a well-mixed, infinitely large population, the system can be described by a pair of ordinary different equations mathematically. The authors study static and dynamic global environments (that is, whether and how omega changes) and obtain some very interesting evolutionary outcomes including the emergence of cyclic evolutions. The findings are supported with both theoretical analysis and numerical approximations (I see no agent-based simulations).

Their work shed light on how diverse dynamics may emerge due to nonlinear interactions between individual performance and environment fluctuations. Nevertheless, I have a few conceptual and mathematical questions that I hope the authors could answer.

(1) The group size N of the public goods game appears in equation (10). How would this parameter affect the evolution of cooperation? (as there is no results from agent-based simulations, at least the authors can do it theoretically/numerically?)

(2) Are there any real-world examples from biology/economy/social science/… that resonate with the findings in the paper? I am particularly interested. After all, the analysis from a dynamical-system perspective may not be intuitive enough.

(3) The captions of the figures can be more self-contained (especially Figure 3-6).

There are also some typos in the manuscript. The authors may do a more careful proofread. To name a few:

Page 4, line 101: an extra ‘the’.

Page 10, line 249-250: ‘max’ and ‘min’ should not be italic.

Besides, I cannot open the URL in Data and Code Availability.

Above all, it is a novel paper and adds to the literature of eco-evolutionary games. I would suggest acceptance with a minor revision.

Reviewer #2: In this paper, the authors explore the effect of the interplay between local and global environmental fluctuations on the evolution of cooperation in a n-player public goods game. Here, local environmental fluctuations are encoded as changes in the productivity factor for cooperators, while global fluctuations determine whether cooperation is promoted or discounted by the environment. In agreement with previous studies, the authors recover periodic behavior of the system in a periodic environment. Interestingly, the periodic solution for chosen parameter values fluctuates in the interior of the (0,1)-interval, while hardly ever going close to either 0 or 1. Another interesting observation is that the system seems to have a continuum of attractors for each value of the environmental functions and once these parameters evolve with time, the system behavior follows these attractors, which is in turn determined by the differences in timescales. Overall, I enjoyed reading this manuscript, but I have a few comments to the authors (please see the report attached).

Reviewer #3: The authors develop a model of an eco-evolutionary game where there is both endogenous environmental feedback and exogenous environmental change. This combination of factors would be applied to strategic settings where there is seasonality or where the game is embedded within a larger system that exerts forcing on the local game-environment dynamics. By combining these exogenous and endogenous drivers of environmental change, the paper makes a nice contribution to evolutionary game theory.

The game that they study is an environmentally dependent public goods game with two relevant ‘environmental’ factors. First is the multiplicative factor of the public good that cooperators experience, which depends on endogenous feedback from the strategies that people use. Second is the curvature of the payoffs as a function of the number of cooperators which can be either concave or convex depending on the state of an exogenously changing global environment. As the global environment fluctuates, there are alternating periods with increasing and decreasing marginal returns to cooperation.

My overarching comment in the manuscript concerns the connections that the authors make between their model its practical implications. Lines 333-343 state that this model shows how complex dynamics can emerge from nonlinear interactions among strategies and the environment. I fully agree with the authors on this claim. However, they also assert that the model has ‘practical value for solving social dilemmas is an ever-changing world’. In the introduction, many systems which exhibit social dilemmas are discussed, however this does not carry through to the remainder of the manuscript. While the authors do at times relate their model to such systems (e.g. crowdsourcing; skilled vs unskilled collaboration; corporate hiring practices), none of these examples are dealt with in sufficient depth to demonstrate how this model could be used to solve social dilemmas in practice. I challenge the authors to develop this aspect of their work, so the manuscript can more clearly demonstrate its theoretical and practical importance.

Comments:

-When the model setup is introduced, it is stated that there is an infinite population that is well mixed, but also that the game has N participants. If I understand correctly, the PGG is played in random groups of N that are drawn from an infinite population. However, I think readers would benefit from the manuscript being more explicit about the assumptions that lead to the payoffs and culminate in equation 6.

-When synergy vs discounting is introduced, it may be more intuitive for many readers to explain how different values of w impact the concavity of P_d(k) and P_c(k). The authors describe these functions as being either ‘synergistic’ or ‘discounting’. A minor comment is that the latter of these terms is often used to describe intertemporal aggregation of payoffs in economics and so it’s use in this paper may cause confusion among some readers.

-Equation 8 defines f(x,r_c), which drives the dynamics of r_c. However, I found the presentation of this equation to be confusing and the text did not help me generate intuition about why this function is used. It would be really helpful if the authors could provide a more intuitive explanation for the why f(x,r_c) takes the form it does, and what it means.

-Figure 2 presents results for the model with a static values of w. It would be great if the figure caption contained enough detail so that a reader could interpret the results shown in the figure without having to refer to the text of the MS. Right now, the caption doesn’t explain that the threshold epsilon* is the minimum value that leads to internal stability, or detail why the blue line in panel b becomes dashed and what this means.

**Have the authors made all data and (if applicable) computational code underlying the findings in their manuscript fully available?**

Reviewer #1: Yes

Reviewer #2: Yes

Reviewer #3: **No: **They list a github repository for their paper. However, it is empty. Therefore, I cannot assess how the figures were made.

PLOS authors have the option to publish the peer review history of their article (what does this mean?). If published, this will include your full peer review and any attached files.

Reviewer #1: **Yes: **Xingru Chen

Reviewer #2: No

Reviewer #3: No
---

## [Decision Letter · Decision Letter 1]

15 May 2023

Dear Mr. Wang,

Thank you very much for submitting your manuscript "Nonlinear eco-evolutionary games with global environmental fluctuations and local environmental feedbacks" for consideration at PLOS Computational Biology. As with all papers reviewed by the journal, your manuscript was reviewed by members of the editorial board and by several independent reviewers. The reviewers appreciated the attention to an important topic. Based on the reviews, we are likely to accept this manuscript for publication, providing that you modify the manuscript according to the review recommendations.

The manuscript has been sent to the same three reviewers who evaluated the original submission. They all agree that the manuscript has improved considerably, and that it can be published, subject to minor modifications. In particular, reviewer #2 makes a number of constructive suggestions that would be worthwhile to consider. In particular, I would like to encourage the authors to carefully proofread their paper.

Regarding the question of whether some parts of the SI should rather go into the main text, I leave the final decision up to the authors (but I believe reviewer #2 provides some good arguments why a presentation in the main text might be preferred).

Sincerely,

Christian Hilbe

Academic Editor

PLOS Computational Biology

James O'Dwyer

Section Editor

PLOS Computational Biology

Reviewer's Responses to Questions

**Comments to the Authors:**

Reviewer #1: The authors have revised their manuscript based on the comments. I would suggest acceptance of the updated version.

Reviewer #2: Please see the report.

Reviewer #3: I thank the authors for their thorough response to the reviews and for meaningfully revising the manuscript to address these comments. The paper makes a meaningful contribution evolutionary game theory by integrating exogenous and endogenous environmental impacts on payoffs and thus dynamics.

**Have the authors made all data and (if applicable) computational code underlying the findings in their manuscript fully available?**

Reviewer #1: Yes

Reviewer #2: None

Reviewer #3: Yes

PLOS authors have the option to publish the peer review history of their article (what does this mean?). If published, this will include your full peer review and any attached files.

Reviewer #1: No

Reviewer #2: No

Reviewer #3: No

Figure Files:

Data Requirements:

Reproducibility:

References:

---

## [Editor Report · Decision Letter 2]

13 Jun 2023

Dear Mr. Wang,

We are pleased to inform you that your manuscript 'Nonlinear eco-evolutionary games with global environmental fluctuations and local environmental feedbacks' has been provisionally accepted for publication in PLOS Computational Biology.

Best regards,

Christian Hilbe

Academic Editor

PLOS Computational Biology

James O'Dwyer

Section Editor

PLOS Computational Biology

The authors have taken the remaining suggestions of reviewer #2 into account. I would like to thank the authors for these changes.

Overall, this is an interesting and valuable manuscript on evolutionary games with environmental feedback, and I'm convinced it will be of interest to the readers of PLoS Computational Biology.

---

## [Editor Report · Acceptance letter]

24 Jun 2023

PCOMPBIOL-D-22-01825R2 

Nonlinear eco-evolutionary games with global environmental fluctuations and local environmental feedbacks

Dear Dr Wang,

I am pleased to inform you that your manuscript has been formally accepted for publication in PLOS Computational Biology. Your manuscript is now with our production department and you will be notified of the publication date in due course.

With kind regards,

Zsofia Freund
